# Use of antisense oligonucleotides to target Notch3 in skeletal cells

**Ernesto Canalis**[1,2,3]*, **Michele Carrer**[4], **Tabitha Eller**[3], **Lauren Schilling**[3], **Jungeun Yu**[1,3]

**1** Department of Orthopaedic Surgery, UConn Health, Farmington, Connecticut, United States of America, **2** Department of Medicine, UConn Health, Farmington, Connecticut, United States of America, **3** The UConn Musculoskeletal Institute, UConn Health, Farmington, Connecticut, United States of America, **4** Ionis Pharmaceuticals, Inc., Carlsbad, California, United States of America

* canalis@uchc.edu

**Data Availability Statement:** All relevant data are within the paper.

**Funding:** This work was supported by the National Institute of Arthritis and Musculoskeletal and Skin Diseases (NIAMS) to EC (Grant AR076747,

## Abstract

Notch receptors are determinants of cell fate and function, and play an important role in the regulation of bone development and skeletal remodeling. Lateral Meningocele Syndrome (LMS) is a monogenic disorder associated with *NOTCH3* pathogenic variants that result in the stabilization of NOTCH3 and a gain-of-function. LMS presents with neurological developmental abnormalities and bone loss. We created a mouse model (*Notch3^{em1Ecan}*) harboring a 6691TAATGA mutation in the *Notch3* locus, and heterozygous *Notch3^{em1Ecan}* mice exhibit cancellous and cortical bone osteopenia. In the present work, we explored whether Notch3 antisense oligonucleotides (ASO) downregulate *Notch3* and have the potential to ameliorate the osteopenia of *Notch3^{em1Ecan}* mice. Notch3 ASOs decreased the expression of *Notch3* wild type and *Notch3^{6691-TAATGA}* mutant mRNA expressed by *Notch3^{em1Ecan}* mice in osteoblast cultures without evidence of cellular toxicity. The effect was specific since ASOs did not downregulate *Notch1*, *Notch2* or *Notch4*. The expression of Notch3 wild type and *Notch3^{6691-TAATGA}* mutant transcripts also was decreased in bone marrow stromal cells and osteocytes following exposure to Notch3 ASOs. *In vivo*, the subcutaneous administration of Notch3 ASOs at 25 to 50 mg/Kg decreased *Notch3* mRNA in the liver, heart and bone. Microcomputed tomography demonstrated that the administration of Notch3 ASOs ameliorates the cortical osteopenia of *Notch3^{em1Ecan}* mice, and ASOs decreased femoral cortical porosity and increased cortical thickness and bone volume. However, the administration of Notch3 ASOs did not ameliorate the cancellous bone osteopenia of *Notch^{em1Ecan}* mice. In conclusion, Notch3 ASOs downregulate *Notch3* expression in skeletal cells and their systemic administration ameliorates cortical osteopenia in *Notch3^{em1Ecan}* mice; as such ASOs may become useful strategies in the management of skeletal diseases affected by Notch gain-of-function.

## Introduction

Notch receptors (Notch1 through 4) are single-pass transmembrane proteins that play a critical role in cell fate and function [1, 2]. *Notch1*, *2* and *3* and low levels of *Notch4* transcripts are

AR072987). M.C. is employed by Ionis Pharmaceuticals, Inc. The content is solely the responsibility of the authors and does not necessarily represent the official view of the National Institutes of Health. The funders of this work had no role in study design, data collection and analysis, decision to publish, or preparation of the manuscript.

**Competing interests:** The authors have declared that no competing interests exist.

**Abbreviations:** The abbreviations used are:ASO, antisense oligonucleotides; BV/TV, bone volume/ total volume; DMEM, Dulbecco's modified Eagle's medium; FBS, fetal bovine serum; HES, Hairy Enhancer of Split; HEY, HES-related with YRPW motif; LMS, Lehman Syndrome or Lateral Meningocele Syndrome; MAML, mastermind; NICD, NOTCH intracellular domain; NRR, negative regulatory region; PEST, proline (P), glutamic acid (E), serine (S) and threonine (T); qRT-PCR, quantitative reverse transcript-polymerase chain reaction; RANKL, receptor activator of nuclear factor- κB ligand; RBPJκ, recombination signal-binding protein for Ig of κ; SMI, structure model index; α-MEM, α-minimum essential medium; μCT, microcomputed tomography.

detected in bone cells, where each receptor acts in a distinct capacity to influence the fate of cells of the osteoblast and osteoclast lineages [3]. Interactions of Notch receptors with ligands of the Jagged and Delta-like families result in the cleavage of NOTCH and the release of its intracellular domain (NICD) [1]. The NICD translocates into the nucleus, and following the formation of a complex with recombination signal-binding protein for Ig of κ (RBPJκ) and mastermind (MAML) it induces the transcription of target genes [4–6]. These include genes encoding Hairy Enhancer of Split (HES)1, 5 and 7 and HES-related with YRPW motif (HEY) 1, 2 and L [7].

Lehman Syndrome or Lateral meningocele syndrome (LMS) (Online Mendelian Inheritance in Man 130720) is a devastating monogenetic disorder associated with pathogenic variants of *NOTCH3* [8–11]. Individuals affected by LMS present with meningoceles, distinct facial features, developmental delay, decreased muscle mass, cardiac valve defects, short stature, scoliosis and bone loss [8–11]. Exome sequencing of families affected by LMS revealed the presence of mutations or deletions in exon 33 of *NOTCH3*, that create a stop codon upstream of sequences coding for the proline (P), glutamic acid (E), serine (S) and threonine (T) (PEST) domain. As a consequence, the PEST domain, which is necessary for the ubiquitination and degradation of NOTCH3, is not translated and NOTCH3 is presumably stable, resulting in a gain-of-function [9]. Autosomal dominant inheritance and de novo heterozygous mutations are reported. Treatment of LMS is not available. NOTCH3 is critical for the function of mural vascular cells and pathogenic variants of *NOTCH3* associated with mutations in the extracellular domain of NOTCH3 cause cerebral autosomal dominant arteriopathy with subcortical infarcts and leukoencephalopathy (CADASIL) [12–15].

In an effort to understand the mechanisms and possible therapeutic avenues to treat individuals with LMS, we created a *Notch3* knock-in mutant mouse model reproducing functional outcomes of the human disease [16]. For this purpose, we introduced a 6691-TAATGA mutation into the mouse genome upstream of sequences coding for the PEST domain, using CRISPR/Cas9 technology to create *Notch3*$^{em1Ecan}$ (syn *Notch3*$^{tm1.1Ecan}$) mutant mice that express a truncated NOTCH3 devoid of the PEST domain.

The administration of antisense oligonucleotides (ASO) has emerged as a novel therapeutic approach to downregulate wild type and mutant transcripts, and has been successful in the silencing of mutant genes in the central and peripheral nervous system, retina, liver and muscle [17–25]. ASOs are single-stranded synthetic nucleic acids that bind target mRNA by Watson-Crick pairing resulting in mRNA degradation by RNase H [26, 27]. Although attempts have been made to transport ASOs to bone, complex delivery systems were necessary and the technology is new to the correction of gene mutations in the skeleton [28].

Approaches to prevent or downregulate Notch signaling include the use of biochemical inhibitors of Notch activation, thapsigargin, antibodies to nicastrin or to Notch receptors or their ligands, and the use of molecules that disrupt the assembly of an active Notch transcriptional complex [29–34]. A limitation of these approaches is that either they are not specific inhibitors of Notch signaling or inhibit all Notch receptors, leading to a generalized Notch knockdown. Antibodies to the negative regulatory region (NRR) of Notch are specific and have been effective at preventing the activation of Notch receptors [35–37]. However, the pronounced downregulation of Notch activation may result in gastrointestinal toxicity, and a possible alternative is the use of second generation ASOs [38].

The purpose of the present work was to determine whether *Notch3* could be downregulated in skeletal cells with a mouse specific Notch3 ASO and possibly ameliorate the osteopenia of *Notch3*$^{em1Ecan}$ mice, a mouse model of LMS and NOTCH3 gain-of-function secondary to the stabilization of the NOTCH3 NICD resulting in higher levels of NOTCH3 activity. We postulated that Notch3 ASOs would target *Notch3* and as a consequence decrease the levels of the

biologically active NOTCH3 NICD. To this end, cells of the osteoblast lineage were obtained from *Notch3^em1Ecan* and control mice and were treated with ASOs targeting *Notch3* to determine their effects on the downregulation of *Notch3*. Notch3 ASOs were tested *in vivo* for their effects on the downregulation of *Notch3* and on the skeletal microarchitecture of heterozygous *Notch3^em1Ecan* mice.

## Materials and methods

### Notch3 antisense oligonucleotides

ASOs targeting *Notch3* mRNA were designed *in silico* by scanning through the entire sequence of murine *Notch3* pre-mRNA, which was screened for potential oligonucleotides complementary to the pre-mRNA. Sequence motifs that were intrinsically problematic because of unfavorable hybridization properties, such as polyG stretches, or potential toxicity due to immunogenic responses, were avoided. Notch3 ASOs were tested for activity *in vitro* for downregulation of *Notch3* mRNA in C2C12 cells at Ionis Pharmaceuticals (Carlsbad, CA), and for activity and toxicity *in vivo* at the Korea Institute of Toxicology (KIT, Daejeon, Korea). For this purpose, 7-week-old *BALB/c* male mice were administered ASOs at a dose of 50 mg/Kg subcutaneously once a week for a total of 3.5 weeks (4 doses). Body weights were determined weekly and mice were euthanized 48 h after the last dose of ASO. Liver, kidney and spleen were weighed, normalized to body weight and compared with organs from control mice. Blood was obtained by cardiac puncture, and plasma was collected for the measurement of alanine aminotransferase, aspartate aminotransferase, total bilirubin, albumin and blood urea nitrogen. These procedures were performed at, and approved by, the Animal Care and Use Committee of the Korea Institute of Toxicology. Total RNA was extracted from lung samples to determine *Notch3* mRNA levels corrected for *cyclophilin A* expression. ASOs that downregulated *Notch3* mRNA in the lung by more than 75% compared to a control ASO and without toxicity *in vivo* were selected. For this study, a mouse Notch3 ASO and control ASO that does not hybridize to any specific mRNA sequence were selected.

### Notch3^em1Ecan mutant mice

A mouse model of Lehman Syndrome, termed *Notch3^em1Ecan* (syn *Notch3^tm1.1Ecan*), harboring a tandem termination at bases 6691–6696 (ACCAAG→TAATGA) in exon 33 of *Notch3* was previously reported and validated [16]. *Notch3^em1Ecan* mice were created in a C57BL/6J background. Genotypes were determined by PCR analysis of tail DNA using forward primer 5′–GTGCTCAGCTTTGGTCTGCTC–3′ and reverse primer 5′–CGCAGGAAGCGCGCTCATTA–3′ for *Notch3^em1Ecan* or 5′–CGCAGGAAGCGGGCCT TGG–3′ for the wild type allele (Integrated DNA Technologies, Coralville, IA). Heterozygous *Notch3^em1Ecan* mutants were crossed with wild type mice to generate ~50% heterozygous *Notch3^em1Ecan* mice and 50% control littermates to be characterized and administered Notch3 ASOs. Studies were approved by the Institutional Animal Care and Use Committee of UConn Health.

### Osteoblast-enriched cell cultures

Osteoblasts were isolated from the parietal bones of 3 to 5 day old control and *Notch3^em1Ecan* mice following exposure to liberase TL 1.2 units/ml (Sigma-Aldrich St. Louis, MO) for 20 min at 37°C for 5 consecutive reactions [39]. The last 3 digestions of cells were pooled and seeded at a density of 10 x 10^4 cells/cm^2, as reported [40]. Osteoblast-enriched cells were cultured in Dulbecco's modified Eagle's medium (DMEM) supplemented with non-essential amino acids (both from Thermo Fisher Scientific, Waltham, MA) and 10% heat-inactivated fetal bovine

serum (FBS; Atlanta Biologicals, Norcross, GA) in a humidified 5% $CO_2$ incubator at 37˚C. Confluent osteoblast-enriched cells were exposed to DMEM supplemented with 10% heat-inactivated FBS, 100 μg/ml ascorbic acid and 5 mM β-glycerophosphate (both from Sigma-Aldrich) in the presence of Notch3 ASO or control ASO at various doses and periods of time as indicated in text and legends.

### Bone marrow stromal cell cultures

Femurs from 4 to 8 week old *Notch3*[em1Ecan] mice and littermate controls were dissected aseptically, the epiphysis removed and bone marrow stromal cells recovered by centrifugation, as described. Cells were pooled and seeded at a density of $1.25 \times 10^6$ cells/cm$^2$ in α-minimum essential medium (α-MEM; Thermo Fisher Scientific) containing heat-inactivated 15% FBS and cultured at 37˚C in a humidified 5% $CO_2$ incubator. At confluence, cells were exposed to α-MEM supplemented with 10% FBS, 100 μg/ml ascorbic acid and 5 mM β-glycerophosphate and cultured in the presence of Notch3 or control ASOs at 20 μM.

### Osteocyte-enriched cultures

To obtain osteocyte-enriched preparations, femurs or tibiae from 4 to 8 week old *Notch3*[em1Ecan] mice and control littermates were collected following sacrifice. Tissues surrounding the bones were dissected, the proximal region to the epiphysis excised and the bone marrow removed by centrifugation. The distal epiphyseal region was removed and the femoral fragments were sequentially exposed for 20 min periods to type II collagenase pretreated with 17 μg/ml Nα-Tosyl-L-lysine chloromethylketone hydrochloride and 5 mM EDTA (Thermo Fisher Scientific) at 37˚C to remove the endosteal and periosteal layers of cells, as described. Osteocyte-enriched bone fragments were obtained and cultured individually in DMEM supplemented with nonessential amino acids, 100 μg/ml ascorbic acid and 10% FBS for 72 h in a humidified 5% $CO_2$ incubator at 37˚C in the presence of Notch3 ASOs or control ASOs at 20 μM.

### Cell proliferation assay

Cell replication was determined using the Cell Counting Kit-8 (*CCK-8*). In this assay, the tetrazolium salt WST-8 [2-(2-methoxy-4-nitrophenyl)-3-(4-nitrophenhyl)-5-(2,4-disulfophenyl)-*2H*-tetrazolium, monosodium salt] produces a formazan dye, measured at an absorbance of 450 nm, upon reduction by cellular dehydrogenases. The assay quantifies viable cells and was used in accordance with manufacturer's instructions (Dojindo Molecular Technologies, Rockville, MD).

### In vivo administration of Notch3 ASOs

Four and a half week old male *Notch3*[em1Ecan] heterozygous mutant and sex-matched wild type littermates were administered Notch3 ASO or control ASO suspended in PBS subcutaneously at various doses and diverse periods of time as indicated in text and legends. To assess an effect on bone microarchitecture, ASOs were administered subcutaneously at a dose of 50 mg/Kg once a week for 4 consecutive weeks to 4.5 week old *Notch3*[em1Ecan] and control mice, and mice were euthanized at 8.5–9 weeks of age.

### Microcomputed tomography (μCT)

Bone microarchitecture of one femur or vertebra (lumbar 3, L3) from experimental and one femur or L3 from control mice was determined using a microcomputed tomography

instrument (μCT 40; Scanco Medical AG, Bassersdorf, Switzerland), which was calibrated periodically using a phantom provided by the manufacturer [41, 42]. Femurs and vertebrae were placed in 70% ethanol and scanned at high resolution, energy level of 55 kVp, intensity of 145 μA and integration time of 200 ms. For cancellous microarchitecture, 160 slices at the distal femoral metaphysis or ~500 slices of L3 were acquired at an isotropic voxel size of 216 μm$^3$ and a slice thickness of 6 μm, and chosen for analysis. Trabecular bone volume fraction and microarchitecture were evaluated starting ~1.0 mm proximal from the femoral condyles. For L3, the vertebral body was scanned in its entirety. Contours were manually drawn a few voxels away from the endocortical boundary every 10 slices to define the region of interest for analysis. The remaining slice contours were iterated automatically. Trabecular regions were assessed for total volume, bone volume, bone volume fraction (bone volume/total volume), trabecular thickness, trabecular number, trabecular separation, connectivity density and structure model index (SMI), using a Gaussian filter (σ = 0.8), and a threshold of 240 permil equivalent to 355.5 mg/cm$^3$ hydroxyapatite [41, 42]. For analysis of femoral cortical bone, contours were iterated across 100 slices along the cortical shell of the femoral midshaft, excluding the marrow cavity. Analysis of bone volume/total volume, porosity, cortical thickness, total cross-sectional and cortical bone area were performed using a Gaussian filter (σ = 0.8, support = 1), and a threshold of 400 permil equivalent to 704.7 mg/cm$^3$ hydroxyapatite.

## Quantitative reverse transcription-polymerase chain reaction (qRT-PCR)

Total RNA was extracted from cells, homogenized tibiae, following the removal of the bone marrow by centrifugation, or osteocyte-enriched fragments with the RNeasy kit or micro-RNeasy Kit (Qiagen, Valencia, CA), in accordance with manufacturer's instructions [43–46]. The integrity of the RNA from tibiae and osteocyte-rich fragments was assessed by microfluidic electrophoresis on an Experion instrument (BioRad, Hercules, CA), and only RNA with a quality indicator number equal to or higher than 7.0 was used for subsequent analysis. Equal amounts of RNA were reverse-transcribed using the iScript RT-PCR kit (BioRad) and amplified in the presence of specific primers (IDT) (Table 1A) with the iQ SYBR Green Supermix or SsoAdvanced Universal SYBR Green Supermix (BioRad) at 60°C for 35 cycles. Transcript copy number was estimated by comparison with a serial dilution of cDNA for *Notch1* (from J. S. Nye, Cambridge, MA), *Notch2* (from Thermo Fisher Scientific) and *Notch4* (from Y. Shirayoshi, Tottori, Japan) [47, 48]. *Notch3* wild type copy number was estimated by comparison to a serial dilution of a 100 to 200 base pair synthetic DNA template (IDT) cloned into pcDNA3.1 (Thermo Fischer Scientific) by isothermal single reaction assembly using commercially available reagents (New England BioLabs, Ipswich, MA) [49].

In experiments where cells or tissues were obtained from *Notch3*$^{em1Ecan}$ and littermate controls, *Notch3* wild type and *Notch3*$^{6691-TAATGA}$ mutant transcripts expressed by heterozygous *Notch3*$^{em1Ecan}$ mice were determined by RT-PCR using fluorescent tagged products. To this end, total RNA was reverse transcribed with Moloney murine leukemia virus reverse transcriptase in the presence of reverse primers of *Notch3* (5'–TGGCATTGGTAGCAGTTC–3') and *Rpl38* (Table 1A). *Notch3* cDNA was amplified by qPCR in the presence of SsoAdvanced Universal Probes Supermix (Bio-Rad) gene expression assay mix, specific Notch3 and Notch3 mutant primers and HEX labeled *Notch3* and FAM labeled *Notch3*$^{6691-TAATGA}$ probes (Table 1B) (Bio-Rad) at 95°C for 10 secs then 60°C for 30 secs and repeated for 45 cycles [50]. *Notch3* or *Notch3*$^{6691-TAATGA}$ mutant transcript copy number was estimated by comparison with a serial dilution of a 100 to 200 bp synthetic DNA fragment (IDT) with or without the 6691TAATGA mutation in the *Notch3* locus cloned into pcDNA3.1(-) [49].

**Table 1. Primers used for qRT-PCR determinations.** GenBank accession numbers identify transcript recognized by primer pairs.

**A. Conventional qRT-PCR**

| Gene | Strand | Sequence | GenBank Accession Number |
|---|---|---|---|
| *Notch1* | Forward | 5'-GTCCCACCCATGACCACTACCCAGTTC-3' | NM_008714 |
| | Reverse | 5'-GGGTGTTGTCCACAGGGGA-3' | |
| *Notch2* | Forward | 5'-TGACGTTGATGAGTGTATCTCCAAGCC-3' | NM_010928 |
| | Reverse | 5'-GTAGCTGCCCTGAGTGTTGTGG-3' | |
| *Notch3* | Forward | 5'-CCGATTCTCCTGTCGTTGTCTCC-3' | NM_008716 |
| | Reverse | 5'-TGAACACAGGGCCTGCTGAC-3' | |
| *Notch4* | Forward | 5'-CCAGCAGACAGACTACGGTGGAC-3' | NM_010929 |
| | Reverse | 5'-GCAGCCAGCATCAAAGGTGT-3' | |
| *Rpl38* | Forward | 5'-AGAACAAGGATAATGTGAAGTTCAAGGTTC-3' | NM_001048057; NM_001048058;NM_023372 |
| | Reverse | 5'-CTGCTTCAGCTTCTCTGCCTTT-3' | |

**B. qRT-PCR using fluorescent tagged PCR products**

| Gene | Strand/Flurophore | Sequence | GenBank Accession Number |
|---|---|---|---|
| *Notch3* | Forward | 5'-AGGACATGGAGAGGAATA-3' | Not Applicable |
| | Reverse | 5'-GGTCAAATAAGGATGCTC-3' | |
| | HEX Fluorophore | 5'-ACCAAGGCCCGCTTCC-3' | |
| *Notch3*[6691-TAATGA] | Forward | 5'-AGGACATGGAGAGGAATA-3' | Not Applicable |
| | Reverse | 5'-GGTCAAATAAGGATGCTC-3' | |
| | FAM Fluorophore | 5'-AGTAGCCCCTAATGAGCGCG-3' | |

## Statistics

Data are expressed as individual sample values, and means ± SD. All data represent biological replicates except for osteoblast-enriched and stromal cell cultures, which represent technical replicates. Quantitative reverse transcript-polymerase chain reaction (qRT-PCR) values were derived from two technical replicates of technical or biological replicates as indicated in the text and legends. Statistical differences were determined by unpaired *t* test for pairwise comparisons or two-way analysis of variance for multiple comparisons with Tukey's post-hoc analysis using GraphPad Prism version 9.3.1 for Mac OS, GraphPad Software (San Diego, CA).

## Results

### Effect of Notch3 ASOs on Notch3 expression in cells of the osteoblast lineage

The effect of Notch3 ASOs was tested in cells of the osteoblast lineage since previous work demonstrated that *Notch3* is preferentially expressed in these cells and is not expressed in cells of the myeloid lineage [16, 51, 52]. Mouse Notch3 ASOs added to the culture medium of osteoblast-enriched cells from C57BL/6 mice at 20 μM decreased *Notch3* mRNA by ~60 to 80% 72 h after ASO addition without microscopic evidence of cellular toxicity or substantial changes in cell replication (Fig 1). The effect of the Notch3 ASO was observed at doses as low as 1 μM and was specific for *Notch3* mRNA since at 20 μM, it did not decrease the expression of *Notch1*, *2* or *4* mRNA (Fig 1). The expression of wild type *Notch3* was lower in osteoblasts from heterozygous *Notch3*[em1Ecan] mice since they carried only one wild type allele, and only *Notch3*[em1Ecan] osteoblasts expressed the *Notch3*[6691TAATGA] transcript (Fig 2). Notch3 ASOs decreased *Notch3* mRNA in osteoblasts and bone marrow stromal cells from wild type and *Notch3*[em1Ecan] mice and *Notch3*[6691TAAGTA] mutant mRNA in *Notch3*[em1Ecan] cells for periods of up to 21 days (Fig 2). The inhibitory effect of Notch3 ASOs was also observed in osteocyte-enriched cells, a

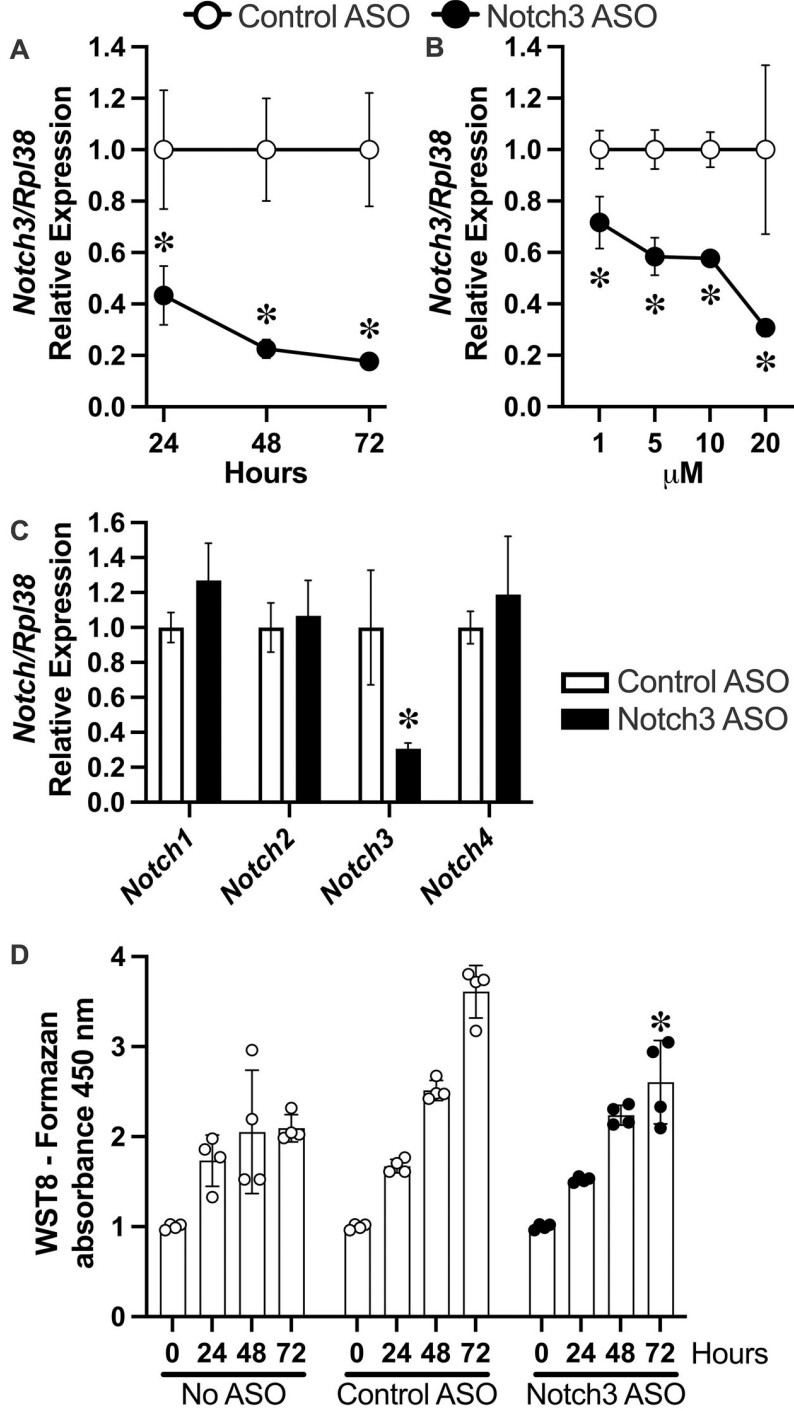

**Fig 1. Effect of control or Notch3 ASOs on *Notch3* mRNA expression and cell replication in calvarial osteoblast-enriched cells.** In panels A and B, *Notch3* mRNA levels were obtained 24 to 72 h (panel A) or 72 h (panel B) after the addition of Notch3 (closed circles) or control ASOs (open circles) at 20 μM in panel A or at 1 to 20 μM in panel B to cells from wild type C57BL/6 mice. In panel C, *Notch1*, *2*, *3* and *4* mRNA levels were obtained 72 h after the addition of Notch3 (black bars) or control ASO (white bars) at 20 μM to cells from wild type C57BL/6 mice. Transcript levels are expressed as relative number following correction for *Rpl38*. Values are means ± SD; n = 3 technical replicates. In panel D, osteoblasts were cultured for 0 to 72 h in the absence or presence of control or Notch3 ASOs at 20 μM and viable cells determined by cell counting kit 8 assay and data expressed as formazan dye and measured at an absorbance of 450 nm. Data are means ± SD; n = 4 technical replicates. *Significantly different between Notch3 ASO and control ASO, $p < 0.05$.

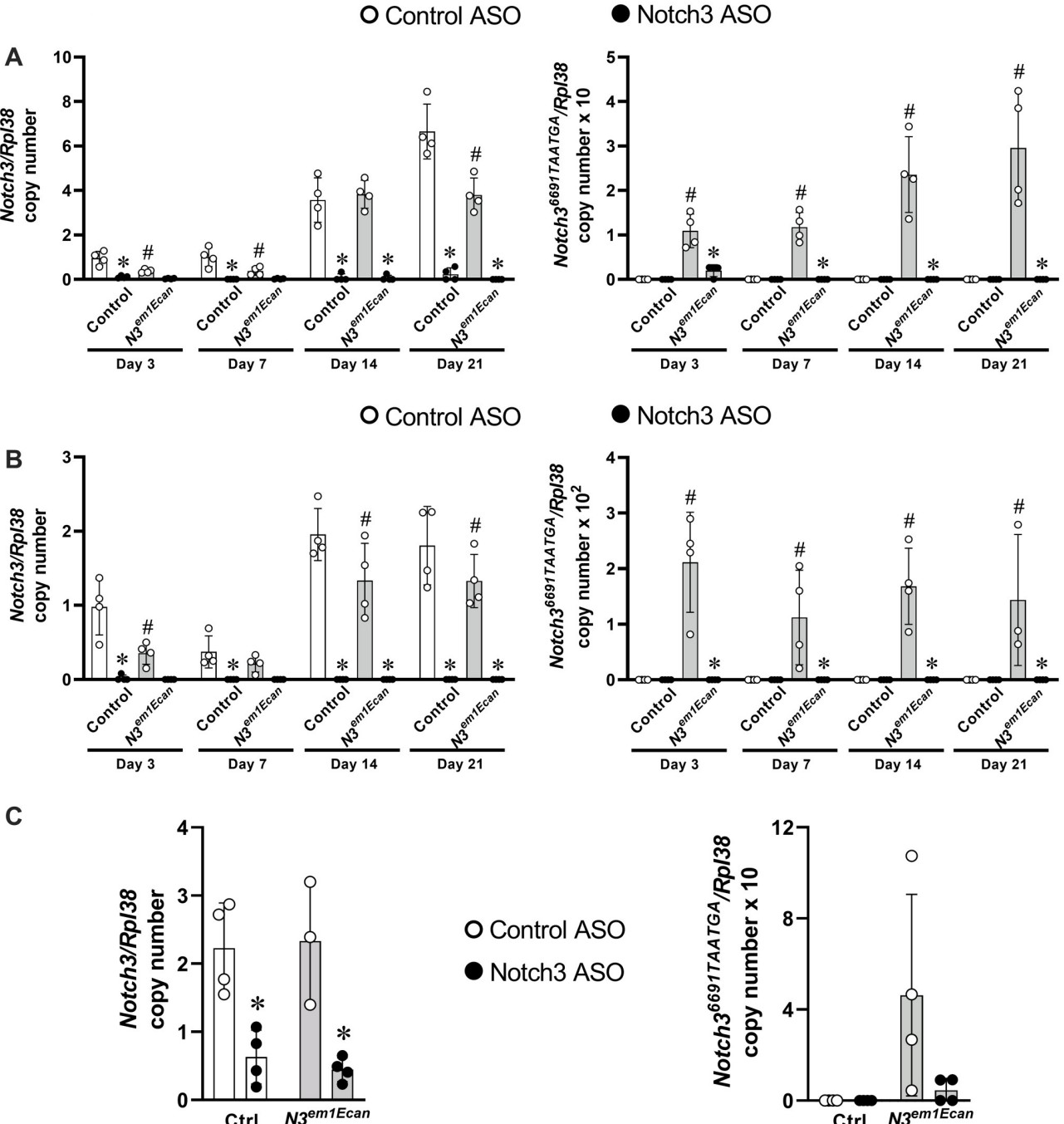

**Fig 2.** Effect of control or Notch3 ASOs on *Notch3* and *Notch3$^{6691TAATGA}$* mutant mRNA in A. calvarial osteoblast-enriched cells, B. bone marrow stromal cells and C. osteocytes from control (white bars) and *Notch3$^{em1Ecan}$* mutant mice (gray bars). Osteoblasts or stromal cells were cultured for 3 weeks and osteocytes for 72h in the presence of Notch3 (closed circles) or control ASOs (open circles) at 20 μM. In A, B and C, transcript levels are expressed as copy number corrected for *Rpl38*. Individual values are shown, and bars and ranges represent means ± SD; n = 4 technical replicates in A and B and biological replicates in C. *Significantly different between Notch3 ASO and control ASO, $p < 0.05$. #Significantly different between *Notch3$^{em1Ecan}$* and control cells, $p < 0.05$.

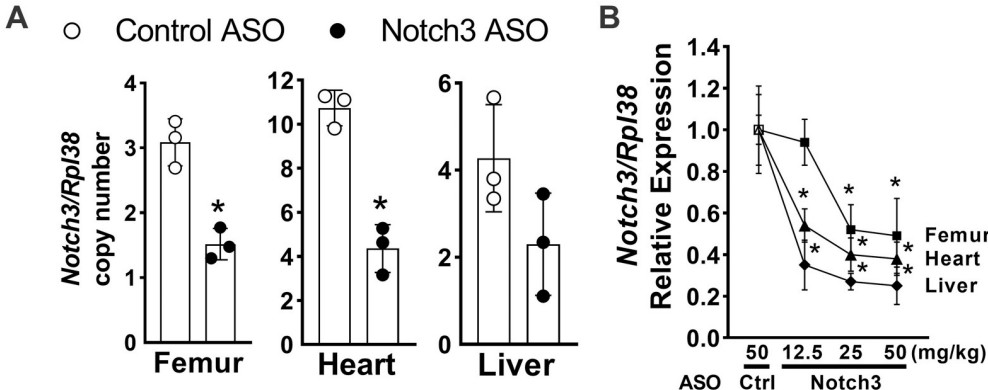

**Fig 3. Effect of control or Notch3 ASOs administered subcutaneously to C57BL/6 wild type mice on *Notch3* mRNA.** In panel A, Notch3 (closed circles) or control (open circles) ASOs were administered at 50 mg/kg and *Notch3* copy number corrected for *Rpl38* was determined 96 h later in femur, heart and liver. In panel B, Notch3 ASOs were administered at the indicated doses and compared to control ASO administered at 50 mg/kg and *Notch3* copy number corrected for *Rpl38* was determined 72 h later in femur, heart, liver and shown as relative expression normalized to a control value of 1. In panel A, individual values are shown, and bars and ranges represent means ± SD. In panel B, values represent means ±SD; n = 3. Data are biological replicates. *Significantly different between Notch3 and control ASO, $p < 0.05$.

cellular environment that preferentially expresses *Notch3* [51, 52]. In this culture system, Notch3 ASOs inhibited *Notch3* expression in wild type and *Notch3^em1Ecan^* mice and also suppressed the expression of *Notch3^6691-TAATGA^* mRNA in *Notch3^em1Ecan^* mutant cells, although this effect did not reach statistical significance (Fig 2).

## Effect of Notch3 ASOs on Notch3 expression *in vivo*

In initial experiments, we tested whether mouse Notch3 ASOs downregulated *Notch3* mRNA *in vivo* in tissues where *Notch3* is expressed. The subcutaneous administration of murine ASOs targeting *Notch3* in C57BL/6 mice at a dose of 50 mg/Kg caused a ~50 to 60% downregulation of *Notch3* mRNA 96 hours later in femur, heart and liver (Fig 3). In a subsequent experiment, Notch3 ASOs, administered subcutaneously to C57BL/6 mice downregulated *Notch3* mRNA 72 hours later in femur when tested at a dose of 25 to 50 mg/Kg, and in heart and liver when administered at doses of 12.5 to 50 mg/Kg (Fig 3).

## Effect of Notch3 ASOs on general characteristics and femoral microarchitecture of Notch3^em1Ecan^ mice

Heterozygous *Notch3^em1Ecan^* mutant male mice were compared to wild type sex-matched littermate controls in a C57BL/6 genetic background. Male mice were studied since they appear to have an osteopenic phenotype that is sustained for up to ~18 weeks of age whereas female *Notch3^em1Ecan^* mutants have osteopenia at 4.5 weeks but not at ~18 weeks of age [16]. Homozygous *Notch3^em1Ecan^* mice were not studied since the homozygous mutation appears to be lethal during embryogenesis [16]. Confirming prior results, *Notch3^em1Ecan^* heterozygous male mice had modest reductions in weight and no changes in femoral length when compared to controls (Fig 4) [16]. Following the administration of mouse Notch3 ASOs, control and *Notch3^em1Ecan^* experimental mice appeared healthy and experienced no changes in weight over a 4 week period. Femoral length was not affected.

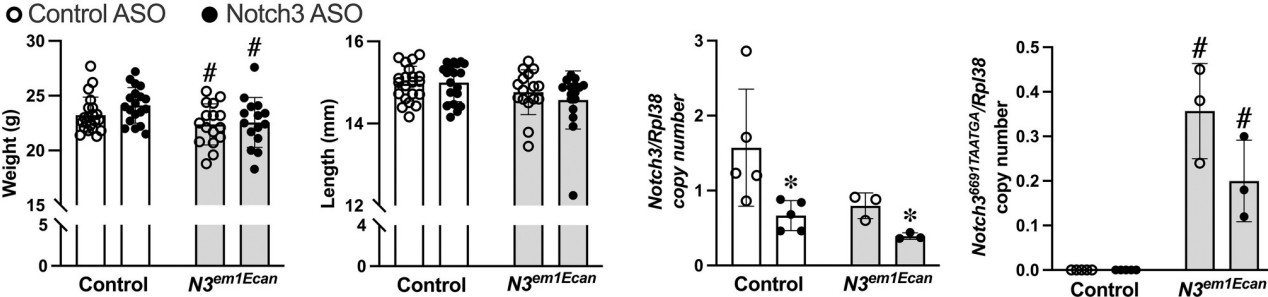

**Fig 4. Body weight, femoral length and *Notch3* transcripts of 8.5–9 week old male *Notch3*$^{em1Ecan}$ mutant mice (gray bars) and littermate controls (white bars) treated with Notch3 ASO (closed circles) or control ASO (open circles) at 50 mg/kg subcutaneously, once a week for 4 weeks.** Individual values are shown, and bars and ranges represent means ± SD; n = 16 to 21 biological replicates, except for mRNA values n = 4 biological replicates. *Significantly different between Notch3 ASO and control, $p < 0.05$. #Significantly different between *Notch3*$^{em1Ecan}$ mutant and control mice, $p < 0.05$.

Notch3 ASOs administered for a 4 week period decreased *Notch3* mRNA expression in wild type mice and in *Notch3*$^{em1Ecan}$ mutant mice. *Notch3*$^{6691-TAATGA}$ mutant mRNA was expressed in tibiae only from *Notch3*$^{em1Ecan}$ mice, and it was suppressed by ~50% following the administration of Notch3 ASOs for 4 weeks, although the effect did not reach statistical significance (Fig 4).

Validating previous observations, μCT of the femoral mid-diaphysis revealed that 8.5–9 week old *Notch3*$^{em1Ecan}$ male mice had decreased cortical bone volume/total volume (BV/TV) associated with increased porosity and decreased cortical thickness (Fig 5) [16]. The subcutaneous administration of mouse Notch3 ASOs once a week at 50 mg/Kg for 4 weeks increased cortical BV/TV and cortical thickness and decreased cortical porosity in *Notch3*$^{em1Ecan}$ mice. A similar, although more modest effect, was noted in wild type mice.

Cancellous bone microarchitecture revealed that *Notch3*$^{em1Ecan}$ mice had decreased cancellous BV/TV and trabecular number when compared to wild type littermates (Fig 6). Administration of Notch3 ASO did not increase femoral (Fig 6) or vertebral cancellous BV/TV significantly in either *Notch3*$^{em1Ecan}$ or control littermates. BV/TV (%) was (means ± SD; n = 18) 20.1 ± 2.8 in control mice treated with control ASO and (n = 17) 20.6 ± 2.7 in control treated with Notch3 ASO. BV/TV (%) was (n = 16) 15.0 ± 2.3 in *Notch3*$^{em1Ecan}$ mice treated with control ASO and (n = 14) 16.4 ± 3.3 in *Notch3*$^{em1Ecan}$ mice treated with Notch3 ASO. Therefore, the effect of Notch3 ASO was limited to the cortical bone osteopenia, possibly because Notch3 is preferentially expressed by the osteocyte and cortical bone is enriched in these cells [51, 52].

## Discussion

The present work demonstrates that Notch3 ASOs downregulate *Notch3* mRNA in cells from wild type and *Notch3*$^{em1Ecan}$ mice, a preclinical model of Lehman Syndrome (LMS). Notch3 ASOs were effective *in vitro* in cells of the osteoblast lineage, where *Notch3* is expressed. The effect was observed in the absence of obvious cellular toxicity although a modest decrease in cell number was noted. Notch3 ASOs were specific since they did not modify the expression of Notch1, 2 and 4 in osteoblast-enriched cells.

Multiple approaches to downregulate Notch signaling have been reported; however, often they are not specific to this signaling pathway or to a particular Notch receptor. A recent alternative has been the use of antibodies to the NRR of individual Notch receptors to prevent

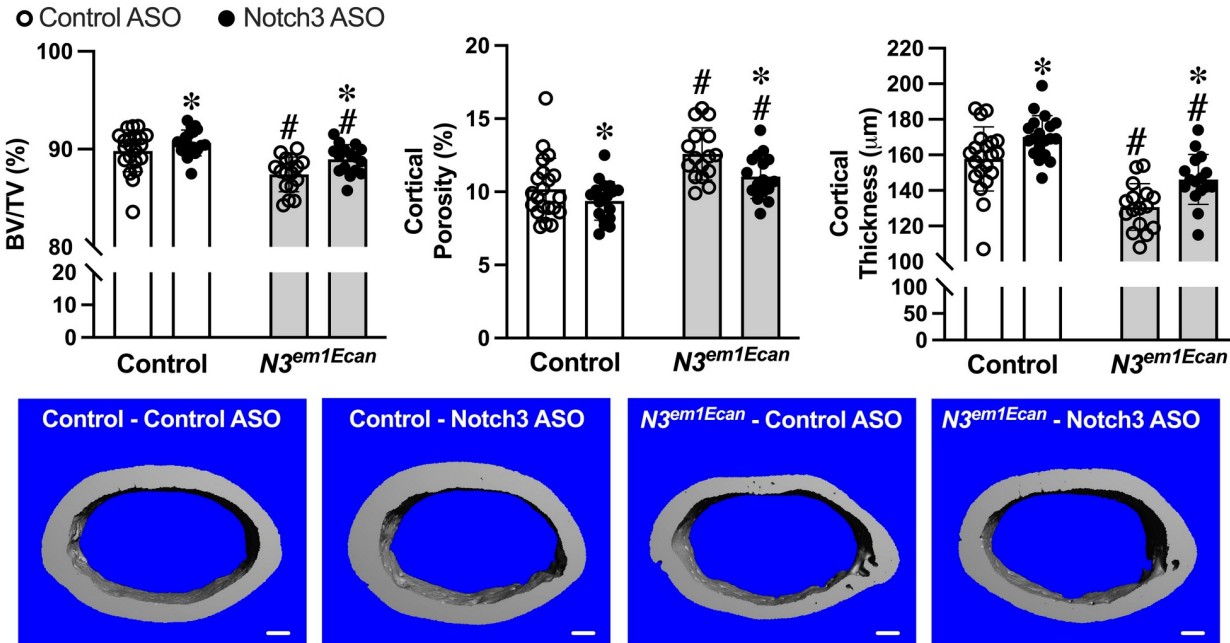

**Fig 5. Cortical bone microarchitecture assessed by μCT of the mid-diaphyseal femur from 8.5–9week old _Notch3_<sup>em1Ecan</sup> mutant (gray bars) male mice and sex-matched littermate controls (white bars) treated with Notch3 ASO (closed circles; n = 19 for control, n = 16 for _Notch3_<sup>em1Ecan</sup>) or control ASO (open circles; n = 21 for control, n = 16 for _Notch3_<sup>em1Ecan</sup>) both at 50 mg/kg subcutaneously, once a week for 4 weeks prior to sacrifice.** Parameters shown are cortical bone volume/total volume (BV/TV, %), cortical porosity (%), and cortical thickness (mm). Individual values are shown, and bars and ranges represent means ± SD of biological replicates. *Significantly different between Notch3 and control ASO, $p < 0.05$. #Significantly different between _Notch3_<sup>em1Ecan</sup> and control, $p < 0.05$. Representative images show cortical bone osteopenia in _Notch3_<sup>em1Ecan</sup> mutant mice and its amelioration by Notch3 ASOs. Scale bars in the right corner represent 100 μm.

Notch activation by blocking the NRR from exposure to the γ-secretase complex [35–37]. We have demonstrated that anti-Notch2 NRR and anti-Notch3 NRR antibodies are effective in reversing the skeletal phenotype of _Notch2_<sup>tm1.1Ecan</sup> and of _Notch3_<sup>em1Ecan</sup> mice, models of Hajdu Cheney Syndrome and LMS, respectively [36, 37]. Although anti-Notch NRR antibodies are specific, the significant downregulation of the Notch receptor could lead to gastrointestinal toxicity.

Alternate approaches to downregulate specific Notch receptors have included ASOs, and we demonstrated that Notch2 ASOs downregulate Notch2 expression _in vitro_ and _in vivo_ and ameliorate the osteopenia of mice harboring a _Notch2_ mutation replicating the one found in Hajdu Cheney Syndrome [38]. Similar to these observations, we expected that Notch3 ASOs would downregulate _Notch3_ wild type and mutant transcripts, and as a consequence decrease the expression of NOTCH3.

In the present study, we confirm that specific Notch ASOs are a suitable alternative to decrease Notch activation in conditions of Notch gain-of-function. However, the effect of Notch3 ASOs was less pronounced than the one reported with anti-Notch3 NRR antibodies [37]. This is possibly because the anti-Notch3 antibody targets the NRR located in the transmembrane domain of NOTCH3, whereas the Notch3 ASO targets _Notch3_ transcripts and as a consequence has to penetrate the cell to be effective. It is also possible that blocking Notch activation is a more effective way to prevent a Notch gain-of-function, or anti-Notch3 antibodies might be more stable or gain better access to target tissues than Notch3 ASOs. Notch3 ASOs were effective at ameliorating the cortical osteopenia of _Notch_<sup>em1Ecan</sup> mice without evidence of

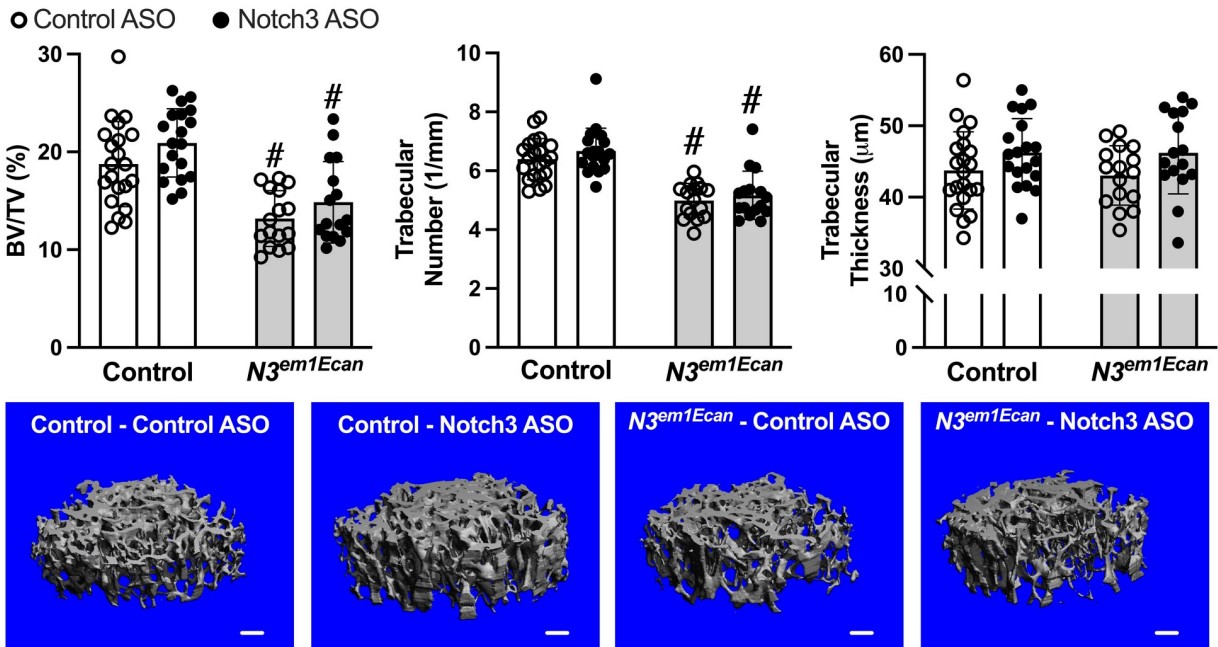

**Fig 6. Cancellous bone microarchitecture assessed by µCT of the distal femur from 8.5–9 week old *Notch3^em1Ecan* mutant (gray bars) male mice and sex-matched littermate controls (white bars) treated with Notch3 ASO (closed circles; n = 19 for mutant, n = 16 for *Notch3^em1Ecan*) or control ASO (open circles; n = 21 for control, n = 16 for *Notch3^em1Ecan*) both at 50 mg/kg subcutaneously, once a week for 4 weeks prior to sacrifice.** Parameters shown are trabecular bone volume/total volume (BV/TV, %); trabecular number (1/mm) and thickness (um). Individual values are shown, and bars and ranges represent means ± SD of biological replicates. #Significantly different between *Notch3^em1Ecan* and control, $p < 0.05$. Representative images show cancellous bone osteopenia in *Notch3^em1Ecan* mutant mice and no effect by Notch3 ASO. Scale bars in the right corner represent 100 µm.

apparent toxicity. However, Notch3 ASOs did not modify the cancellous bone osteopenia of *Notch3^em1Ecan* mice at either femoral or vertebral sites. The effect of Notch3 ASOs was not explored at other skeletal sites since µCT parameters of bone microarchitecture are only established for cortical and cancellous bone [41]. It is possible that Notch3 ASOs were more effective at the osteocyte-rich cortical compartment because *Notch3* is preferentially expressed by these cells [11, 51–53]. Although the work demonstrates that *Notch3* and *Notch3^6691-TAATGA* mutant mRNA were downregulated in osteocyte-rich preparations, we did not demonstrate that Notch3 ASOs are more effective in this cell population following their administration *in vivo*. It is possible that Notch3 ASOs were transported more efficiently to cortical than to cancellous bone. This may be related to differences in the blood vessel network between cortical and cancellous bone allowing for better access of the systemically administered Notch3 ASO to the cortical compartment. Blood supply of cortical long bones is carried out by a central nutrient artery (2/3) and periosteal arteries (1/3), whereas the metaphyseal, cancellous-rich bone, is irrigated by the epiphyseal arteries [54, 55]. However, perfusion efficiency does not appear to be different between epiphyseal and diaphyseal bone [56].

Attempts have been made to enhance the transport of ASOs to bone using complex delivery systems, and the technology has not been applied to the correction of gene mutations in the skeleton [28]. In this study, a practical systemic approach was used to downregulate Notch3 in skeletal and non-skeletal tissue by the subcutaneous administration of Notch3 ASOs. We demonstrate that a murine Notch3 ASO downregulated *Notch3* in tissues where the gene is expressed, including bone. The decrease in *Notch3* in a mouse model of LMS was associated

with partial recovery of bone mass at cortical sites. Although this was not complete, a significant effect on cortical BV/TV and thickness was achieved with amelioration of the *Notch3*[em1Ecan] cortical phenotype.

The present findings confirm that a mouse model replicating a mutation found in LMS displays femoral cancellous and cortical bone osteopenia [16]. The phenotype of the *Notch3*[em1Ecan] mutant mouse recapitulates aspects of LMS including osteopenia, but not the neurological manifestations of the disease [11, 16]. The osteopenic phenotype is manifested early in life in mice of both sexes, although it seems to be more persistent in male mice and for this reason we elected to study and treat 4.5 week old male *Notch3*[em1Ecan] mice in an attempt to ameliorate the osteopenic phenotype. Because only male mice were reported, it is important not to extrapolate the results to female mice. Phenotypic alterations of experimental and control mice were assessed by μCT, and analyses required the *ex vivo* exam of bone following the sacrifice of mice. Therefore, the same animal could not be analyzed before and after the administration of Notch3 ASOs.

The osteopenia of *Notch3*[em1Ecan] mutants has been attributed to an increase in osteoclast number and bone resorption secondary to an increased expression of receptor activator of nuclear factor- κB ligand (RANKL) by cells of the osteoblast lineage [57]. *Notch3* is not expressed in the myeloid lineage; and therefore, NOTCH3 does not have direct effects on osteoclastogenesis [3, 11, 16]. Consequently, direct effects of Notch3 ASOs in this lineage were not examined.

## Conclusions

In conclusion, Notch3 ASOs downregulate *Notch3* expression in skeletal cells from a mouse model of LMS and ameliorate its cortical osteopenic phenotype. Consequently, ASOs may become a useful strategy in the management of skeletal diseases affected by gain-of-Notch function.

## Acknowledgments

The authors thank Magda Mocarska for technical assistance, Mary Yurczak for secretarial support, Christopher Bonin and Genevieve Hargis for artwork and creation of figures.

## Author Contributions

**Conceptualization:** Ernesto Canalis.

**Formal analysis:** Ernesto Canalis.

**Funding acquisition:** Ernesto Canalis.

**Investigation:** Tabitha Eller, Lauren Schilling, Jungeun Yu.

**Methodology:** Ernesto Canalis, Michele Carrer.

**Project administration:** Ernesto Canalis.

**Resources:** Michele Carrer.

**Supervision:** Ernesto Canalis.

**Validation:** Ernesto Canalis, Tabitha Eller, Lauren Schilling, Jungeun Yu.

**Visualization:** Jungeun Yu.

**Writing – original draft:** Ernesto Canalis.

**Writing – review & editing:** Ernesto Canalis, Michele Carrer, Tabitha Eller, Lauren Schilling, Jungeun Yu.

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
