## [Decision Letter · Decision Letter 0]

30 Mar 2022

PONE-D-22-04880Use of antisense oligonucleotides to target Notch3 in skeletal cellsPLOS ONE

Dear Dr. Canalis,

Thank you for submitting your manuscript to PLOS ONE. After careful consideration, we feel that it has merit but does not fully meet PLOS ONE’s publication criteria as it currently stands. Therefore, we invite you to submit a revised version of the manuscript that addresses the points raised during the review process.

We look forward to receiving your revised manuscript.

Kind regards,

Gary Stein

Academic Editor

PLOS ONE

Journal Requirements:

Reviewers' comments:

Reviewer's Responses to Questions

**Comments to the Author**

1. Is the manuscript technically sound, and do the data support the conclusions?

Reviewer #1: Yes

Reviewer #2: Yes

2. Has the statistical analysis been performed appropriately and rigorously? 

Reviewer #1: Yes

Reviewer #2: Yes

3. Have the authors made all data underlying the findings in their manuscript fully available?

Reviewer #1: Yes

Reviewer #2: Yes

4. Is the manuscript presented in an intelligible fashion and written in standard English?

Reviewer #1: Yes

Reviewer #2: Yes

5. Review Comments to the Author

Reviewer #1: The authors created a mouse model for Lateral Meningocele Syndrome (LMS), a very rare disease associated with gain-of-function of the Notch3 Receptor. The mouse model showed cancellous and cortical bone osteopenia. Then, the Notch3 receptor was downregulated by weekly injections of antisense oligonucleotides (ASO) specific for Notch3. The effect on the expression of Notch3 was examined by RT-PCR. Interestingly, the LMS-phenotype in bone was partially reversed. The effect of the AOS was stronger in cortical than in cancellous bone. However, the LMS-phenotype was not as extensively reversed with ASO in comparison to the use of a Notch3 antibody. In 2020, the authors published the same mouse model, but used an antibody instead of ASO to reverse the gain-of-function effect in the LMS mouse. Thus, the paper is presenting a proof-of-principle rather than a significant conceptual advance. The observed differences between cortical and cancellous bone might be related to the higher biological availability of ASO in cortical than cancellous bone. Since a difference is seen, the question remains why the authors only examined the femora of the mice. I would a expect a stronger effect in vertebrae.

All in all, the methods are adequate, the conclusions are supported by the data, and the paper is clearly written. However, I have some minor concerns which should be addressed by the authors first.

General:

-Refer to the mice age in weeks all the time. In several sections, months are used for age indication

-17 of 53 citations are self-citations. Many of them are not necessary, especially in the method´s section of the paper. Please, diversify your references.

Introduction:

-The introduction section is sufficient. What about Notch3 loss-of-function? CADASIL? I recommend mentioning it in one or two sentences to give the reader an overall understanding about the function of Notch3 in pathophysiology of humans.

Methods:

-Why are genetic backgrounds of the mice mixed (BALB and C57/BL)?

-Why are different housekeeping genes used as references for Notch3 expression (cyclophilin A and Rpl38?

-Please remove unnecessary references in the method´s section of the paper. Reference does not need to be made for standardized procedures. For example, Ref 38, 39 (page 6) and 40,41 (page 7)

-Were both femora examined in micro-ct or only one side (page 7)

Results:

-Usually, female mice show a more distinctive phenotype than male mice. The authors state that this was different in the Notch3-LMS mouse and give reference to their own paper.

-What is the explanation for this exception? This is a feature of the mouse model that could easily be examined, if male and female mice were analyzed.

-Could the different effect of ASO on cortical vs. cancellous bone be associated with blood perfusion of these areas? The ASO is i.p. injected. Is there data of its distribution available? Fluorescence imaging? If not, please also comment in the discussion section.

Discussion:

-Please, clarify what you had expected from the ASO in LMS-mice and what was different to the anti-Notch3 AB

-When a difference is seen in cortical and cancellous bone in the femora, the effect on calvaria or spine would be interesting. Please comment why this analysis was not performed.

Reviewer #2: This is an interesting report from a group that has worked for some time on the roles of Notch receptors in the physiology of bone formation and resorption and skeletal diseases. This study focuses on Notch 3 and the authors created a Notch3 knock-in mutant mouse to model Lehman Syndrome or Lateral meningocele syndrome (LMS). The gain-of-function mutation results in a truncated NOTCH3 devoid of the PEST domain, which is necessary for ubiquitination and degradation of NOTCH3 to limit its activity. The objective of the study, therefore, appears to be to assess antisense oligonucleotide (ASO) against Notch3 as a potential therapeutic for osteopenia in LMS in Notch3 knock-in mice and in bone cells derived from them. The experiments are well designed but stop short of addressing bone-specific molecular mechanisms and the authors need to better highlight the novel aspects of this work differentiating Notch3 effects from other Notch receptors. Specific suggestions are listed below.

1. In the Abstract the nomenclature for the mouse strains is difficult to follow and this persists throughout the manuscript. The authors state that they created a novel mouse strain“ (Notch3em1Ecan) harboring a 6691-TAATGA mutation in the Notch3 locus, and heterozygous Notch3em1Ecan mice exhibit cancellous and cortical bone osteopenia”. Then later they refer to Notch36691-TAATGA mRNA. This was confusing until I read through the manuscript. Evidently, the heterozygous Notch3em1Ecan was used throughout the experiments because the mice homozygous for the 6691-TAATGA mutation are developmentally abnormal. Thus, the heterozygous Notch3 knock-in mutant have levels of Notch 3 activity that are sufficient to enable demonstration of amelioration of osteopenia with Notch3 ASOs. Please consider more consistent use of nomenclature here and throughout the manuscript.

2. The concluding statement of the Abstract simply restates the findings without providing a conclusion about important insights resulting from the study.

3. The last paragraph of the Introduction also needs some attention regarding the hypothesis to be test and concluding insights. One thing that needs highlighting is that the work is modeling LMS, where Notch3 is abnormally activated, so treatment Notch3 ASOs could normalize bone status by inhibitingNotch3 signaling. Is this interpretation correct?

4. Page 6, line 2: This sentence does not seem correct: “Heterozygous Notch3em1Ecan mutants were crossed with wild type mice to create heterozygous Notch3em1Ecan”.

5. Page 14: Regarding the results in Figure 6, there is an assumption made, based on previously published work, that the preferential effect of Notch3 ASO was limited to cortical bone osteopenia, without increasing BV/TV in cancellous bone, because Notch3 is expressed preferentially by osteocytes. Showing these effects in cortical versus cancellous bone in the mouse tissues after ASO treatment would enhance the mechanistic value of the work.

6. Page 15: Similarly in the Discussion, direct evidence that the Notch3 ASOs were more effective in the osteocyte-rich compartment in Notch3 knock-in mutants, accounting for the preferential targeting of cortical bone, would add value to the study.

7. Page 17: As in the Abstract, this single sentence does not constitute a conclusion.

6. PLOS authors have the option to publish the peer review history of their article (what does this mean?). If published, this will include your full peer review and any attached files.

Reviewer #1: No

Reviewer #2: No

---

## [Author Response · Author response to Decision Letter 0]

21 Apr 2022

April 18, 2022

Gary Stein

Academic Editor

PLOS ONE

Dear Dr. Stein:

In accordance with your letter of March 30, 2022, we would like to submit a revised version of the manuscript entitled “Use of antisense oligonucleotides to target Notch3 in skeletal cells” (PONE-D-22-04880) to be considered for publication by PLOS One. Changes are outlined in this response and highlighted in the text of the paper. 

Reply to Reviewer #1: The authors created a mouse model for Lateral Meningocele Syndrome (LMS), a very rare disease associated with gain-of-function of the Notch3 Receptor. The mouse model showed cancellous and cortical bone osteopenia. Then, the Notch3 receptor was downregulated by weekly injections of antisense oligonucleotides (ASO) specific for Notch3. The effect on the expression of Notch3 was examined by RT-PCR. Interestingly, the LMS-phenotype in bone was partially reversed. The effect of the AOS was stronger in cortical than in cancellous bone. However, the LMS-phenotype was not as extensively reversed with ASO in comparison to the use of a Notch3 antibody. In 2020, the authors published the same mouse model, but used an antibody instead of ASO to reverse the gain-of-function effect in the LMS mouse. Thus, the paper is presenting a proof-of-principle rather than a significant conceptual advance. The observed differences between cortical and cancellous bone might be related to the higher biological availability of ASO in cortical than cancellous bone. Since a difference is seen, the question remains why the authors only examined the femora of the mice. I would a expect a stronger effect in vertebrae.

All in all, the methods are adequate, the conclusions are supported by the data, and the paper is clearly written. However, I have some minor concerns which should be addressed by the authors first.

General:

-Refer to the mice age in weeks all the time. In several sections, months are used for age indication

 As recommended, we now refer to the age of mice in weeks throughout the paper. 

-17 of 53 citations are self-citations. Many of them are not necessary, especially in the method´s section of the paper. Please, diversify your references.

 Unnecessary references 38, 39, 40 and 41 were removed. Added 10 additional references from other laboratories. 

Introduction:

-The introduction section is sufficient. What about Notch3 loss-of-function? CADASIL? I recommend mentioning it in one or two sentences to give the reader an overall understanding about the function of Notch3 in pathophysiology of humans.

 We now mention CADASIL in the Introduction and cite references on the function of NOTCH3 in mural vascular cells and role of NOTCH3 in CADASIL. Please see page 3, end of paragraph 2.

Methods:

-Why are genetic backgrounds of the mice mixed (BALB and C57/BL)?

 The initial screen and toxicology studies of ASOs were performed at the Korea Institute of Toxicology in BALB/c mice. The ASOs that were active and had no toxicity were chosen for the present studies. All the work presented in this paper was conducted in Notch3em1Ecan and littermate controls in a C57BL/6 background.

-Why are different housekeeping genes used as references for Notch3 expression (cyclophilin A and Rpl38?

 The initial screen of activity of the ASOs was conducted at the Korea Institute of Toxicology and transcript levels corrected for Cyclophilin A. The ASOs downregulating Notch3 mRNA >75% were selected for the work presented in this paper. All mRNA data shown in this paper were generated in the Canalis laboratory and are corrected for Rpl38.

-Please remove unnecessary references in the method´s section of the paper. Reference does not need to be made for standardized procedures. For example, Ref 38, 39 (page 6) and 40,41 (page 7)

 References 38, 39, 40 and 41 were removed as requested.

-Were both femora examined in micro-ct or only one side (page 7)

 Only one side, and this is now stated in page 7, last paragraph.

Results:

-Usually, female mice show a more distinctive phenotype than male mice. The authors state that this was different in the Notch3-LMS mouse and give reference to their own paper. What is the explanation for this exception? This is a feature of the mouse model that could easily be examined, if male and female mice were analyzed.

 The reviewer is correct and we apologize for the error. We examined the published paper (Canalis, et al. J. Biol Chem, 2018) and Notch3em1Ecan female mice have more pronounced osteopenia at 1 month of age than male mice. The error was corrected. The reason why male mice were chosen to test the effect of the ASOs is because the osteopenic phenotype is more persistent and consistent in Notch3em1Ecan male than in female mice. At 4 months of age, male but not female mice are osteopenic. In this study, 1 month old male mice were administered ASOs and examined at 2 months of age to ensure that a phenotype would be detected. The statement was changed to indicate that male mice were studied because the osteopenic phenotype in males was sustained for up to 18 weeks. Please see page 13, paragraph 2. 

-Could the different effect of ASO on cortical vs. cancellous bone be associated with blood perfusion of these areas? The ASO is i.p. injected. Is there data of its distribution available? Fluorescence imaging? If not, please also comment in the discussion section.

 We reviewed information on the vascular system supplying long bones as well as perfusion studies. Blood supply for the cortex of long bones is carried out by a central nutrient artery and periosteal arteries whereas metaphyseal-epiphyseal arteries irrigate the bone metaphysis rich in cancellous bone. Therefore, it is possible that Notch3 ASOs had greater access to cortical than cancellous bone. However, perfusion studies have not shown differences in perfusion efficiency in epiphysis, diaphysis or calvariae. We now present this information in the Discussion and quote appropriate references. Please see page 17, end of paragraph 1. 

Discussion:

-Please, clarify what you had expected from the ASO in LMS-mice and what was different to the anti-Notch3 AB

 We now explain in the Discussion that we expected a downregulation of Notch3 expression by the Notch3 ASOs. We also provide possible explanation for the greater effect of anti-Notch3 antibodies such as greater stability or access to target tissues. It is possible that the anti-Notch3 antibody is more effective than the ASO because it targets the transmembrane domain of NOTCH3 and does not need to enter the cell to be effective. The ASO instead acts intracellularly to target the mRNA for degradation. Please see page 16, last paragraph.

-When a difference is seen in cortical and cancellous bone in the femora, the effect on calvaria or spine would be interesting. Please comment why this analysis was not performed.

 We analyzed the effect of Notch3 ASOs in cancellous bone at L3 in wild type and Notch3em1Ecan mice and results were analogous to those observed in femoral cancellous bone. A statement to this effect was included in the Results and data for L3 BV/TV were included in the text. Please see page 14, paragraph 2.

 Calvariae were not examined since parameters of microarchitectural analysis of bone (the primary analysis utilized in this study) are established for cancellous and cortical bone present in vertebrae and long bones and not for intramembranous bone present in calvariae (Bouxsein et al., J Bone Miner Res, 25:1468-1486, 2010).

Reply to Reviewer #2: This is an interesting report from a group that has worked for some time on the roles of Notch receptors in the physiology of bone formation and resorption and skeletal diseases. This study focuses on Notch 3 and the authors created a Notch3 knock-in mutant mouse to model Lehman Syndrome or Lateral meningocele syndrome (LMS). The gain-of-function mutation results in a truncated NOTCH3 devoid of the PEST domain, which is necessary for ubiquitination and degradation of NOTCH3 to limit its activity. The objective of the study, therefore, appears to be to assess antisense oligonucleotide (ASO) against Notch3 as a potential therapeutic for osteopenia in LMS in Notch3 knock-in mice and in bone cells derived from them. The experiments are well designed but stop short of addressing bone-specific molecular mechanisms and the authors need to better highlight the novel aspects of this work differentiating Notch3 effects from other Notch receptors. Specific suggestions are listed below.

1. In the Abstract the nomenclature for the mouse strains is difficult to follow and this persists throughout the manuscript. The authors state that they created a novel mouse strain“ (Notch3em1Ecan) harboring a 6691-TAATGA mutation in the Notch3 locus, and heterozygous Notch3em1Ecan mice exhibit cancellous and cortical bone osteopenia”. Then later they refer to Notch36691-TAATGA mRNA. This was confusing until I read through the manuscript. Evidently, the heterozygous Notch3em1Ecan was used throughout the experiments because the mice homozygous for the 6691-TAATGA mutation are developmentally abnormal. Thus, the heterozygous Notch3 knock-in mutant have levels of Notch 3 activity that are sufficient to enable demonstration of amelioration of osteopenia with Notch3 ASOs. Please consider more consistent use of nomenclature here and throughout the manuscript.

 We now qualify Notch36691-TAATGA as Notch36691-TAATGA mutant RNA expressed by the Notch3em1Ecan mutant mouse. In the Abstract and throughout the manuscript, we qualify the Notch36691-TAATGA as a mutant transcript and Notch3em1Ecan as the mouse model expressing the transcript.

2. The concluding statement of the Abstract simply restates the findings without providing a conclusion about important insights resulting from the study.

 In accordance with the reviewer’s comment, a more appropriate conclusion was added in the Abstract and Conclusions. 

3. The last paragraph of the Introduction also needs some attention regarding the hypothesis to be test and concluding insights. One thing that needs highlighting is that the work is modeling LMS, where Notch3 is abnormally activated, so treatment Notch3 ASOs could normalize bone status by inhibitingNotch3 signaling. Is this interpretation correct?

 We now highlight that Notch3em1Ecan is a model of LMS and of gain-of-NOTCH3 function, and we postulate that by decreasing Notch3 mRNA expression with specific ASOs one could decrease the levels of biologically active NOTCH3 NICD. Please see page 4, last paragraph.

 Strictly speaking, ASOs downregulate transcripts. By decreasing Notch3 mRNA, there is less active NOTCH3 NICD and less signaling.

4. Page 6, line 2: This sentence does not seem correct: “Heterozygous Notch3em1Ecan mutants were crossed with wild type mice to create heterozygous Notch3em1Ecan”.

 The sentence was corrected. It now states, Heterozygous Notch3em1Ecan mutants were crossed with wild type mice to generate ~50% heterozygous Notch3em1Ecan mice and 50% control littermates to be characterized and administered Notch3 ASOs. Please see page 6, paragraph 1.

5. Page 14: Regarding the results in Figure 6, there is an assumption made, based on previously published work, that the preferential effect of Notch3 ASO was limited to cortical bone osteopenia, without increasing BV/TV in cancellous bone, because Notch3 is expressed preferentially by osteocytes. Showing these effects in cortical versus cancellous bone in the mouse tissues after ASO treatment would enhance the mechanistic value of the work.

 The reviewer is correct, and we assume that the Notch3 ASO was more effective in osteocyte-rich cortical bone. Notch 3 is preferentially expressed by the osteocyte as previously published by us and others (Zanotti et al., Bone, 103:159, 2017 and Delgado-Calle et al., Caner Res, 76:1089, 2016). However, as the reviewer indicates, we cannot state with certainty that this is the reason why the Notch3 ASO was more effective in cortical bone. Other possibilities, such as differential vascularization of cortical and cancellous bone as pointed out by reviewer 1, could play a role and are now included in the Discussion. Please see page 17, paragraph 1. We also indicate that we do not demonstrate that Notch3 ASOs are more effective in an osteocyte population in vivo. The experiment proposed by the reviewer would be valuable but technically quite challenging since one would need to show downregulation of Notch3 and Notch3 mutant mRNA in different bone compartments possibly by in situ hybridization and the method would not be sufficiently sensitive to detect modest differences or reductions in transcript expression. We sincerely apologize for this deficiency and state the deficiency in the Discussion. Please see page 17, paragraph 1.

6. Page 15: Similarly in the Discussion, direct evidence that the Notch3 ASOs were more effective in the osteocyte-rich compartment in Notch3 knock-in mutants, accounting for the preferential targeting of cortical bone, would add value to the study.

 We agree with the reviewer; but as mentioned in the response to comment 5, we believe that it would be technically difficult to demonstrate a downregulation in bone tissue.

7. Page 17: As in the Abstract, this single sentence does not constitute a conclusion. 

 We have modified the conclusion statement – “In conclusion, Notch3 ASOs downregulate Notch3 expression in skeletal cells from a mouse model of LMS and ameliorate its cortical osteopenic phenotype. Consequently, ASOs may become a useful strategy in the management of skeletal diseases affected by gain-of-Notch function”.

Additional Modification

 As we were correcting the manuscript, we noticed some inconsistencies in the copy number of “transcripts” shown in Figure 2. To ensure absolute accuracy, we repeated the real time PCR of all the samples shown in Figure 2A, B and C and the figure was replaced with the newly generated transcript data. Rest assured that the effect of Notch3 ASO remains the same and the only change is in the value of the actual copy number.

We hope you now find the manuscript acceptable for publication. Thank you for your consideration.

Sincerely yours, 

Ernesto Canalis, M.D.

Director, Center for Skeletal Research

Professor of Orthopaedics and Medicine

---

## [Editor Report · Decision Letter 1]

26 Apr 2022

Use of antisense oligonucleotides to target Notch3 in skeletal cells

PONE-D-22-04880R1

Dear Dr. Canalis,

We’re pleased to inform you that your manuscript has been judged scientifically suitable for publication and will be formally accepted for publication once it meets all outstanding technical requirements.

Kind regards,

Gary Stein

Academic Editor

PLOS ONE

Additional Editor Comments (optional):

Concerns raised by reviewers were adequately addressed so the manuscript is acceptable for publication
---

## [Editor Report · Acceptance letter]

29 Apr 2022

PONE-D-22-04880R1 

Use of antisense oligonucleotides to target Notch3 in skeletal cells 

Dear Dr. Canalis:

I'm pleased to inform you that your manuscript has been deemed suitable for publication in PLOS ONE. Congratulations! Your manuscript is now with our production department. 

Kind regards, 

on behalf of

Dr. Gary Stein 

Academic Editor

PLOS ONE